# OccVAR: Scalable 4D Occupancy World Model via Next-Scale Prediction

## Abstract

In this paper, we propose OCCVAR, a generative occupancy world model that simulates the movement of the ego vehicle and the evolution of the surrounding environment. Different from visual generation, the occupancy world model should capture the fine-grained 3D geometry and dynamic evolution of the 3D scenes, posing great challenges for the generative models. Recent approaches based on autoregression (AR) have demonstrated the potential to predict vehicle movement and future occupancy scenes simultaneously from historical observations, but they typically suffer from the **inefficiency** and **temporal degradation** in long-time generation. To holistically address the efficiency and quality issues, we propose a spatial-temporal transformer via temporal next-scale prediction, aiming at predicting the 4D occupancy scenes from coarse to fine scales. To model the dynamic evolution of the scene, we incorporate the ego movement before the tokenized occupancy sequence, enabling the prediction of ego movement and controllable scene generation. To model the fine-grained 3D geometry, OCCVAR utilizes a muitli-scale scene tokenizer to capture the hierarchical information of the 3D scene. Experiments show that OCCVAR is capable of high-quality occupancy reconstruction, long-time generation and fast inference speed compared to prior works.

## 1 Introduction

Recent years have seen great advancements in the development of autonomous driving (AD) systems. While existing AD methods (Hu et al., 2023b; Jiang et al., 2023; Hu et al., 2022; Huang et al., 2023; Yang et al., 2024) have demonstrated excellent results across a range of driving scenarios, there are still challenges when dealing with long-tail distributions or out-of-distribution situations. A promising direction to address these challenges is world models, which simulate and comprehend the surrounding environment by learning a comprehensive representation of the external world.

Occupancy world model, as a specialized type of world model, has gained significant attention for its expressiveness of the 3D geometry. The occupancy world model takes the historical occupancy observations and movement of ego car as input, aiming at forecasting the future 3D scene evolutions and planning a safe trajectory of the ego car. Different from vision-based approaches, the 3D occupancy can describe the fine-grained 3D structure of the scene, demonstrating superior expressiveness of the 3D road scenes. Several occupancy world models (Zheng et al., 2023; Wei et al., 2024; Wang et al., 2024a) have been developed in recent years. Despite their compelling results, these methods suffer from two limitations: **inefficiency** and **temporal degradation**, especially in long-time generation.

There are two potential approaches to construct the occupancy world model: diffusion-based methods and autoregressive methods. Diffusion-based techniques (Wang et al., 2024a; Liu et al., 2023) leverage a diffusion model to generate occupancy scenes. However, these methods face the problem of **inefficiency** due to the multiple denoising steps involved in inference.

In contrast, GPT-style autoregressive methods (Zheng et al., 2023; Wei et al., 2024) generate occupancy scenes sequentially in an autoregressive manner. The GPT framework (Radford et al., 2019) has been successfully applied to image/video generation (Yu et al., 2022; Esser et al., 2021; Yan et al., 2021; Kondratyuk et al., 2023) and demonstrates superior generation ability. Following the

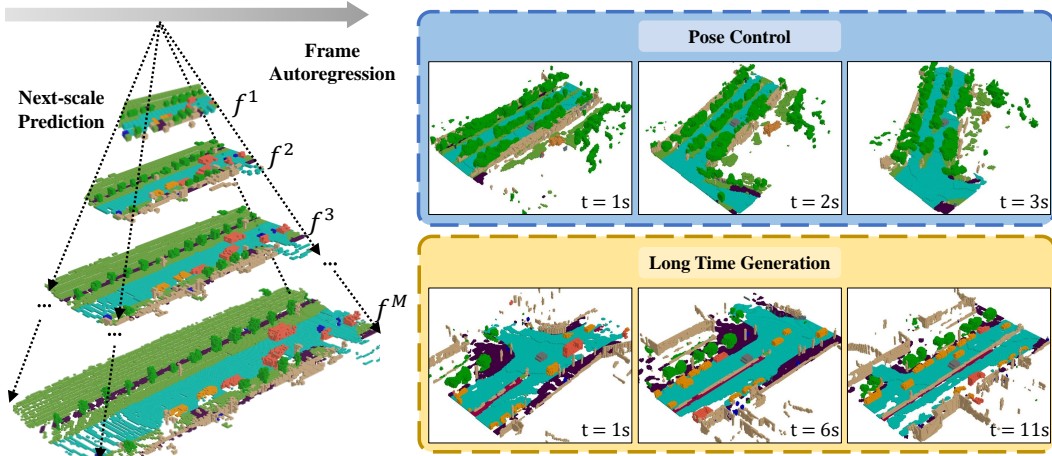

Figure 1: We propose OCCVAR, a novel autoregressive occupancy world model that generates occupancy scenes from coarse to fine scales (lower to higher resolutions). Our model is capable of generating long-time occupancy scenes (right) in short times, demonstrating the effectiveness and efficiency of the proposed framework.

GPT structure, a naive implementation for occupancy world model is to utilize a VQVAE-like tokenizer to represent the scene as discrete tokens, which are then flattened into a one-dimensional sequence and input to transformers for next-token prediction training. However, this vanilla approach, where each generation step depends on the previous one, is also **inefficient**. For example, assume that a scene is comprised of $(50 \times 50)$ discrete tokens. The number of autoregressive steps in next-token prediction for generating 10 frames is 25k, incurring significant inference time costs. Moreover, the flattening process after quantization causes the degradation in the spatial locality and temporal consistency. This results in the **low fidelity** of the generated 4D occupancy scenes, especially for **long-time generation**.

More recently, VAR (Tian et al., 2024a) introduces next-scale prediction (NSP) to handle the inefficiency and spatial degradation problem of next-token prediction (NTP) in image generation, shedding some light on the autoregressive modeling paradigm in occupancy prediction. Nevertheless, different from 2D image generation, the 4D occupancy forecasting requires the model to depict the *fine-grained 3D geometry* of the 3D scenes. Moreover, to generate temporal sequence of occupancy, the model should also be able to capture the dynamics of the scene evolution, which requires a consideration for temporal modeling in next-scale prediction.

In this paper, we propose OCCVAR: a novel autoregressive occupancy world model that is capable of forecasting the future 3D scene evolutions and conducting motion planning for ego vehicle. Specially, we introduce a coarse-to-fine autoregressive modeling mechanism: temporal next-scale prediction (TNSP), inspired from VAR (Tian et al., 2024a), which effectively handles the inefficiency and spatial degradation problem in previous works. To enable the long-time generation and ensure the temporal consistency of the generated occupancy scenes, we develop a novel transformer architecture designed for temporal next-scale prediction. We also develop a multi-scale occupancy tokenizer to capture the hierarchical information of the 3D scenes. The extensive experiments on the nuScenes dataset (Caesar et al., 2020) validate the effectiveness and efficiency of OCCVAR.

Our contributions are summarized as follows:

- We introduce OCCVAR, a coarse-to-fine occupancy world model that incorporates temporal next-scale prediction for autoregressive modeling. A well-designed transformer architecture is specially designed for the temporal modeling of next-scale prediction.

- We propose a multi-scale tokenizer to effectively extract and discretize the hierarchical information of the 3D occupancy scenes.

- OCCVAR outperforms baseline methods by 2.13% on IoU and 1.86% on mIoU and sets state-of-the-art on nuScenes dataset. We also conduct a comprehensive ablation study to demonstrate the effectiveness and efficiency of OCCVAR.

## 2 RELATED WORK

### 2.1 WORLD MODEL

World model (Ha & Schmidhuber, 2018) can predict the consequences of various actions, which is crucial for autonomous driving. Traditional models emphasize visual prediction (Hu et al., 2023a; Zhao et al., 2024; Su et al., 2024; Gao et al., 2024; Wang et al., 2023; Lu et al., 2023; Wang et al., 2024b; Zheng et al., 2024; Gao et al., 2024; Jiang et al., 2024), potentially overlooking the essential 3D information needed for AD vehicles. Some approaches attempt to forecast point clouds using unannotated LiDAR scans (Zhang et al., 2023; Zyrianov et al., 2024), but these methods neglect semantic information and are not suitable for vision-based or fusion-based autonomous driving. Occupancy world models (Zheng et al., 2023; Wei et al., 2024; Wang et al., 2024a) create a world model in 3D occupancy space, providing a more comprehensive understanding of the evolvement of 3D scenes.

### 2.2 VISUAL AUTO-REGRESSIVE GENERATION

Visual Autoregressive (AR) Generation refers to utilizing autoregressive methods to generate images (Esser et al., 2021; Razavi et al., 2019; Lee et al., 2022; Yu et al., 2021; 2022) or videos (Yan et al., 2021; Kalchbrenner et al., 2017; Yu et al., 2023). Generally, the AR models employ a raster-scan paradigm, which encodes and flattens 2D images into 1D token sequences. Recently, VAR (Tian et al., 2024a) proposes to utilize next-scale prediction in visual auto-regressive modeling, which effectively handle the inefficiency and spatial degradation of next-token prediction. However, it still remains unclear whether the next-scale prediction is suitable for occupancy prediction. Moreover, the modeling process of VAR does not take accounts for the temporal dimension, which still requires a consideration for how to adapt this modeling approach for 4D occupancy generation, which is the main focus of our work.

## 3 METHOD

In this work, we propose **OCCVAR**, a novel occupancy world model designed to comprehend historical observations and forecast the future 3D scenarios. As illustrated in Fig. 2, our proposed OCCVAR consists of two components: a robust tokenizer that encodes 3D occupancy and ego motion into discrete tokens (see Sec. 3.1), and a generative world model using next-scale prediction for future 3D scene forecasting and motion planning (see Sec. 3.2).

### 3.1 TOKENIZER

The tokenizer aims to model the 3D occupancy scene and the ego motion as discrete tokens. To achieve this, our tokenizer consists of two components: scene tokenizer and motion tokenizer.

#### 3.1.1 SCENE TOKENIZER

The goal of scene tokenizer is to model the 3D occupancy scene as discrete tokens. To achieve this, a common practice for scene tokenizer is to employ a quantized autoencoder like VQVAE (Zheng et al., 2023; Wei et al., 2024), which quantizes the occupancy feature map with discrete feature vectors. However, unlike natural language sentences with an inherent left-to-right ordering, the occupancy feature map are inter-dependent, resulting in the bidirectional correlations of the quantized token sequence. This contradicts the unidirectional dependency assumption of autoregressive models, where each token can only depend on its prefix, as illustrated in Tian et al. (2024a). Thus, we propose a multi-scale tokenizer specifically designed for next-scale prediction.

**Occupancy Encoder.** Firstly we employ an occupancy encoder to encode the occupancy scene into a BEV feature map. Given a scene $\mathbf{S} \in \mathbb{R}^{H_{raw} \times W_{raw} \times D_{raw}}$ with $L$ semantic classes, where $H_{raw}$, $W_{raw}$, $D_{raw}$ represents the resolution of the 3D volume, we first convert it to a BEV representation. We employ an embedding layer to embed the 3D occupancy scene into a latent space. Then we convert the 3D scene to a BEV representation by merging the height dimension with the channel dimension. We utilize a series of 2D convolution layers to compress the BEV map to a latent feature

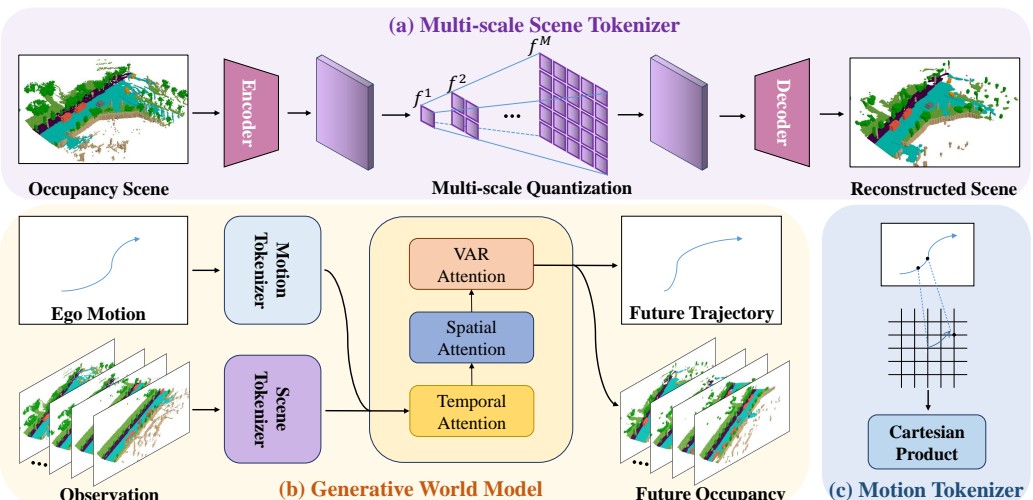

Figure 2: **Overview of OCCVAR.** OCCVAR consists of two components: a robust tokenizer (a and c) that encodes 3D occupancy and ego motion into discrete tokens, and a generative world model (b) using next-scale prediction for future 3D scene forecasting and motion planning.

$\mathbf{F} \in \mathbb{R}^{H \times W \times C}$, where $H = H_{raw}/d$, $W_d = W_{raw}/d$ and $d$ is a hyper-parameter that determines the down-sampling factor.

**Multi-scale Quantizer.** The quantizer aims to tokenize the latent feature $\mathbf{F}$ into multi-scale discrete tokens. Previous approaches (Zheng et al., 2023; Wei et al., 2024) attempt to transform the feature into a collection of codebook entries through vector quantization, where each entry is responsible for a small area. However, operating tokenization solely on local information may result in the loss of global context. Inspired from visual generation works (Tian et al., 2024a), we develop a multi-scale quantizer to discretize the occupancy feature into multi-scale discrete tokens.

Assume that we want to quantize the latent feature $\mathbf{F} \in \mathbb{R}^{H \times W \times C}$ to $M$ multi-scale discrete token maps: $\mathbf{F}_q = (\mathbf{f}_q^1, \mathbf{f}_q^2, ..., \mathbf{f}_q^M)$, each with resolution $(h_1, w_1, h_2, w_2, ..., h_M, Y_M)$. Note that $h_M = H$ and $w_m = W$. We define a learnable codebook $\mathbf{C} \in \mathbb{R}^{V \times C}$, consisting of $V$ vectors, each embedding with a dimension of $C$.

The multi-scale quantization is conducted from low resolution to high resolution. To get the low resolution features at $m$-th scale, we utilize an interpolation function to downsample the latent BEV feature $\mathbf{F} \in \mathbb{R}^{H \times W \times C}$ to the low resolution $(h_m, w_m)$, resulting in $\mathbf{f}^m \in \mathbb{R}^{h_m \times w_m \times C}$. Then we conduct quantization by replacing the vectors in the interpolated feature map $\mathbf{f}^m$ to its nearest neighbour look-up in the codebook $\mathbf{C}$. This process can be formulated as:

$$\mathbf{f}_q^m = \arg\min_{\mathbf{q} \in \mathbf{C}} \|\mathbf{q} - \mathbf{f}^m\|_2, \quad \mathbf{f}^m = \mathcal{I}(\mathbf{f}, (x_m, y_m)), \tag{1}$$

where $\mathcal{I}$ is the interpolation function. In this way, we get the quantized tokens $\mathbf{f}_q^m$ at the first scale.

With a residual design, the interpolated feature map $\mathbf{f}$ would be updated after getting the quantized tokens of each scale. For example, after getting $\mathbf{f}_q^m$ at $m$-th scale, we would upsample the $\mathbf{f}_q^m$ to the original resolution $(H, W)$, which is then subtracted from the feature map $\mathbf{F}$. This process can be formulated as:

$$\mathbf{F}' = \mathbf{F} - \phi_m(\mathcal{I}(\mathbf{f}_q^m, (H, W))), \tag{2}$$

where $\mathcal{I}$ is the interpolation function and $\phi_m$ represents the convolution layers to address the information loss in upsampling.

The whole process would be repeated until we get all of the $M$ scale tokens $\mathbf{F}_q = (\mathbf{f}_q^1, \mathbf{f}_q^2, ..., \mathbf{f}_q^M)$. The quantized multi-scale tokens would be passed as input to the subsequent reconstruction and generation modules.

**Occupancy Decoder.** The occupancy decoder takes the multi-scale quantized tokens as input and output the reconstructed 3D occupancy scenes. Firstly, we convert the multi-scale quantized tokens $\mathbf{F}_q = (\mathbf{f}_q^1, \mathbf{f}_q^2, ..., \mathbf{f}_q^M)$ to the BEV feature map $\hat{\mathbf{F}}$. We upsample the features of each scale to the original resolution and then pass them as input to the convolution layers $\phi_{1,...,M}$, which is the same as those in 2. The interpolated results are accumulated to get the reconstructed bev feature map. This process can be formulated as:

$$\hat{\mathbf{F}} = \sum_{m=1}^{M} \phi_m(\mathcal{I}(\mathbf{f}_q^m, (x_d, y_d))) \tag{3}$$

To reconstruct the 3D occupancy scene from $\hat{\mathbf{F}} \in \mathbb{R}^{X_d \times Y_d \times C}$, we utilize another series of convolution layers to upsample the BEV feature map $\hat{\mathbf{F}}$ to resolution $X \times Y$, and then split the height dimension from the channel dimension. Finally we apply an MLP to transform the channel dimension to the class number for classification, resulting in $\hat{\mathbf{S}}_{raw} \in \mathbb{R}^{X \times Y \times Z \times L}$, where $L$ is the class number.

### 3.1.2 MOTION TOKENIZER

The motion tokenizer is utilized to discretize the motion of ego vehicle for better integrating it into our sequence prediction model. We utilize the position $x, y$ and orientation $\theta$ relative to the previous frame to represent the motion of the vehicle. We discard the information in the z-axis because the vehicle' speed in z-axis is nearly zero in most time. We apply a vanilla uniform quantization of the motion information, resulting in $V_x$, $V_y$ and $V_\theta$ tokens in vocabulary. Then we map the relative motion with three discrete tokens to a motion token $\mathbf{P}$ by Cartesian product:

$$\mathbf{P} = \mathcal{E}(x + y \times V_x + \theta \times V_x \times V_y), \tag{4}$$

where $\mathcal{E}$ is an embedding layer. By doing so, we can account for the vehicle's expected trajectory and control how it influences the occupancy predictions, ensuring that the model captures how the environment evolves as the vehicle moves.

### 3.2 GENERATIVE WORLD MODEL

In this section, we introduce our generative world model via next-token prediction, as shown in Fig. 2. As **input**, we assume the tokenized BEV features $\mathbf{F} = \{\mathbf{F}_1, ..., \mathbf{F}_{T-1}\}$ and ego motion $\mathbf{P} = \{\mathbf{P}_1, ..., \mathbf{P}_{T-1}\}$ with previous $T - 1$ frames. The **target output** of the world model is the occupancy scene $\mathbf{S}_T$ and motion $\mathbf{P}_T$ at the $T$-th frame.

### 3.2.1 PRELIMINARY

Some previous occupancy world models (Wei et al., 2024) utilize a vanilla next-token autoregressive modeling on occupancy prediction. Considering the occupancy map with resolution $(n \times n)$, the likelihood of the sequence $x = \{x_1, x_2, ..., x_{n \times n}\}$ can be decomposed to the product of $n \times n$ conditional probabilities:

$$p(x) = \prod_{i=1}^{n \times n} p\left(x_t \mid x_1, x_2, \ldots, x_{i-1}\right) \tag{5}$$

However, such next-token prediction introduces several issues:

(1) *Inefficiency*. With a vast number of occupancy tokens to generate, producing each token sequentially in an autoregressive manner results in a significant computational cost. The complexity of generating a occupancy feature map with resolution $(n \times n)$ is $O(n^6)$ (see Proof at Appendix. A.1).

(2) *Structural degradation*. After quantization, the occupancy tokens are flatten for next-token autoregressive modeling, which disrupts the spatial and temporal locality inherent in the occupancy feature map.

A naive solution for the issues is to utilize next-scale prediction Tian et al. (2024a), which have demonstrated successful practice in image generation. This modeling approach can reduce the generating complexity to $O(n^4)$ (see Proof at Appendix. A.1). Moreover, there is no flattening operation

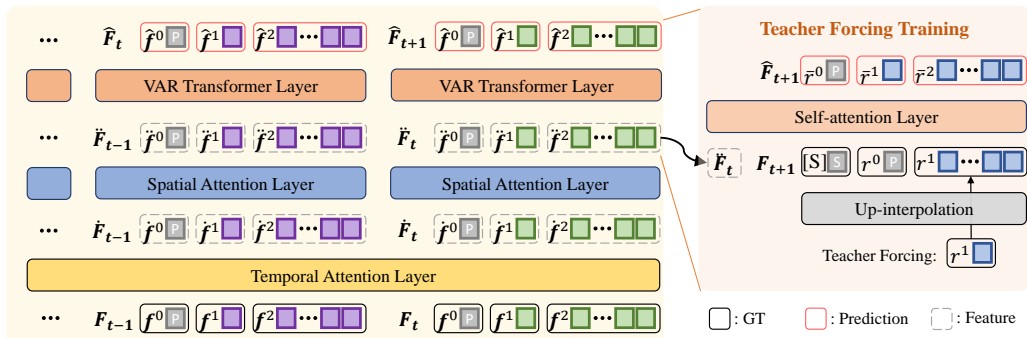

Figure 3: **Transformer block design.** We employ three types of attention blocks: temporal attention blocks, spatial attention blocks and VAR generation blocks.

in next-scale prediction, and tokens in the same scale are fully correlated. This multi-scale design would not disrupt the spatial locality in the occupancy feature map. However, how to model the temporal dependency of the occupancy sequence and how to maintain the temporal consistency of the generated frames still remains unexplored.

### 3.2.2 TRANSFORMER BLOCK DESIGN

We propose a well-designed transformer to adapt the next-scale autoregressive modeling approach for temporal generation, details shown in Fig. 3. To better integrate the ego-vehicle motion into our prediction paradigm, we treat the motion token as 0-th scale token $\mathbf{f}_0$ and splice it before the multi-scale occupancy tokens, resulting in $M + 1$ tokens $\{\mathbf{f}_0, \mathbf{f}_t^1, ..., \mathbf{f}_t^M\}$ for each frame. We employ three types of attention blocks: temporal attention blocks, spatial attention blocks and VAR attention blocks, as shown in Fig. 3.

The *temporal attention blocks* aim to model the temporal dependencies of the occupancy frames. In this process, we utilize a block-wise causal attention mask to ensure that each token $\mathbf{f}_t^m$ can only attend to its prefix and the tokens in the same scale: $\{\mathbf{F}_1, \mathbf{F}_2, \ldots, \mathbf{F}_{t-1}, \mathbf{f}_t^1, \mathbf{f}_t^2, \ldots, \mathbf{f}_t^{m-1}\}$. After temporal attention blocks, we denote the token of $t$-th frame at $m$-th scale as $\dot{\mathbf{f}}_t^m$

Then we utilize *spatial attention blocks* to capture the spatial relationship of different scales. In this process, the attention scores are computed within a scene. We utilize an intra-frame full-attention mask to ensure that each token $\dot{\mathbf{f}}_t^m$ has a global view at the whole scene $\{\dot{\mathbf{f}}_t^1, \dot{\mathbf{f}}_t^2, \ldots, \dot{\mathbf{f}}_t^M\}$. After spatial attention blocks, the token of $t$-th frame at $m$-th scale as $\ddot{\mathbf{f}}_t^m$.

Finally, we utilize *VAR attention blocks* to forecast the tokens in the next frame. During training period, when generating the tokens at frame $t + 1$, the model takes the features of its previous frame $\ddot{\mathbf{F}}_t$ and the start token [S] as input. When generating the $m$-th scale at $t$-th frame, the autoregressive likelihood is formulated as:

$$p(x) = \prod_{m=0}^{M} p(\hat{\mathbf{f}}_{t+1}^m \mid \ddot{\mathbf{F}}_t, \hat{\mathbf{f}}_{t+1}^0, \hat{\mathbf{f}}_{t+1}^1, \ldots, \hat{\mathbf{f}}_{t+1}^{m-1}) \tag{6}$$

where the $\hat{\mathbf{f}}_t^0$ is the motion token and $\hat{\mathbf{f}}_t^m$ denotes the $k$-th scale tokens at $t$-th frame. Note that we utilize a block-wise causal attention mask to ensure that each token $\hat{\mathbf{f}}_{t+1}^m$ can only attend to its prefix $\{\ddot{\mathbf{F}}_t, \hat{\mathbf{f}}_{t+1}^0, \hat{\mathbf{f}}_{t+1}^1, \ldots, \hat{\mathbf{f}}_{t+1}^{m-1}\}$. To better demonstrate the design of each block, we show the visualization of attention mask of different attention blocks in the Appendix. A.2.

### 3.3 LOSS FUNCTION

When training the scene tokenizer, we utilize cross-entropy loss and lovasz-softmax loss (Berman et al., 2018). To enhance the the global occupancy reconstruction performance, we also utilize

| Method | Input | MIOU(%)↑ | | | | IOU(%)↑ | | | |
|---|---|---|---|---|---|---|---|---|---|
| | | 1s | 2s | 3s | Avg. | 1s | 2s | 3s | Avg. |
| OccWorld-F (Zheng et al., 2023) | Cam | 8.03 | 6.91 | 3.54 | 6.16 | 23.62 | 18.13 | 15.22 | 18.99 |
| OccLLaMA-F (Wei et al., 2024) | Cam | 10.34 | 8.66 | 6.98 | 8.66 | 25.81 | 23.19 | 19.97 | 22.99 |
| **OCCVAR-F (Ours)** | Cam | **17.17** | **10.38** | **7.82** | **11.79** | **27.60** | **25.14** | **20.33** | **24.35** |
| OccWorld-O (Zheng et al., 2023) | Occ | 25.78 | 15.14 | 10.51 | 17.14 | 34.63 | 25.07 | 20.18 | 26.63 |
| OccLLaMA-O (Wei et al., 2024) | Occ | 25.05 | 19.49 | 15.26 | 19.93 | 34.56 | 28.53 | 24.41 | 29.17 |
| **OCCVAR-O (Ours)** | Occ | **27.96** | **21.75** | **16.47** | **22.06** | **38.73** | **29.50** | **24.86** | **31.03** |

• The "-O" represents the results utilizing ground truth occupancy as input.

• The "-F" represents that the input is multi-view camera images.

Table 1: Quantitative results of 4D occupancy forecasting. The "-O" represents the results utilizing ground truth occupancy as input. The "-F" represents that the input is multi-view camera images and we use FBOCC (Li et al., 2023) to predict the occupancy from images. We can see that our OCCVAR outperforms previous work in a large margin.

geoscal loss and semscal loss illustrated in Cao & De Charette (2022), which optimize the class-wise derivable precision, recall and specificity for semantics and geometry. In general, our loss function is defined as: $\mathcal{L} = \lambda_1 \mathcal{L}_{ce} + \lambda_2 \mathcal{L}_{lovasz} + \lambda_3 \mathcal{L}_{geoscal} + \lambda_4 \mathcal{L}_{semscal}$, where the factors $\lambda_{1,2,3,4}$ are used to balance the losses.

When training the world model, we utilize cross-entropy loss for the generation of occupancy tokens and pose tokens. The loss function is defined as: $\mathcal{L} = \beta_1 \mathcal{L}_{occ} + \beta_2 \mathcal{L}_{pose}$.

## 4 EXPERIMENTS

### 4.1 EXPERIMENTAL SETUP

**Datasets and Metrics.** We evaluate our method on the nuScenes (Caesar et al., 2020) dataset. The nuScenes dataset is collected in Boston and Singapore and comprised of 1000 driving sequences. We inherit the official nuScenes (Caesar et al., 2020) split setting for evaluation, where the train/val scenes are 700 and 150, respectively. Each sequence lasts for around 20 seconds and the key-frames are annotated at 2 Hz. We employ the occupancy annotation in Occ3D (Tian et al., 2024b) based on nuScenes.

Following common practices, we utilize 2-second historical context (4 frames) and forecast the subsequent 3-second scenes (6 frames) unless specified. We report the mIoU and IoU for the 4D occupancy prediction task.

**Implementation Details.** Our training period consists of 2 stages: tokenization and generation. For tokenization, we downsample the occupancy with a factor of 8. The codebook are comprised of 4096 nodes and the channel dimension of the codebook entry is 128. We utilize 6 scales with [1,5,10,15,20,25] for multi-scale settings. In the tokenizer loss function, the $\lambda_1, \lambda_2, \lambda_3, \lambda_4$ are 10.0, 1.0, 0.3, 0.5 respectively. For generation, we utilize 4 layers each for three blocks of our methods. The hidden dimension and head number are 128 and 4, respectively. The $\beta_1$ and $\beta_2$ are 1.0 and 1.0 respectively. More details could be found in Appendix. A.3.

### 4.2 MAIN RESULTS

Our generative world model can accomplish two tasks: 4D occupancy forecasting task and motion planning task. The 4D occupancy forecasting task aims to forecast the future observation of 3D occupancy scene. The motion planning task aims to calculate a sequence of trajectory points for the ego vehicle.

**4D Occupancy Forecasting** In this experiment, we compare our method with state-of-the-art approaches on the 4D occupancy forecasting task. Following common practice, we conduct our evaluation in two settings: (1) using ground-truth 3D occupancy data (-O); and (2) using predicted results from FBOCC Li et al. (2023) based on camera data (-F). The results are shown in Tab. 1. We observe that our OCCVAR achieves significant performance gain over existing methods in short time forecasting within 3 seconds. Specifically, for occupancy input, we can see that compared with Oc-

| Method | Input | Supervision | L2(m)↓ | | | | Coll.(%)↓ | | | |
|--------|-------|-------------|------|------|------|------|------|------|------|------|
| | | | 1s | 2s | 3s | Avg. | 1s | 2s | 3s | Avg. |
| ST-P3 (Hu et al., 2022) | Cam | M. & B. & D. | 1.33 | 2.11 | 2.90 | 2.11 | 0.23 | 0.62 | 1.27 | 0.71 |
| UniAD (Hu et al., 2023b) | Cam | M. & B. & Mot. & T. & Occ. | 0.48 | **0.96** | **1.65** | **1.03** | 0.05 | **0.17** | **0.71** | **0.31** |
| VAD† (Jiang et al., 2023) | Cam | M. & B. & Mot. | 0.54 | 1.15 | 1.98 | 1.22 | 0.04 | 0.39 | 1.17 | 0.53 |
| OccWorld-F | Cam | Occ. | 0.45 | 1.33 | 2.25 | 1.34 | 0.08 | 0.42 | 1.71 | 0.73 |
| **OCCVAR-F (Ours)** | Cam | Occ. | 0.46 | 1.37 | 2.23 | 1.35 | 0.15 | 0.47 | 1.89 | 0.83 |
| OccNet (Tong et al., 2023) | Occ | M. & B. | 1.29 | 2.31 | 2.98 | 2.25 | 0.20 | 0.56 | 1.30 | 0.69 |
| OccWorld-O | Occ | None | **0.43** | 1.08 | 1.99 | 1.17 | 0.07 | 0.38 | 1.35 | 0.60 |
| **OCCVAR-O (Ours)** | Occ | None | 0.45 | 1.10 | 2.02 | 1.21 | 0.12 | 0.42 | 1.80 | 0.78 |

• M., B., D., Mot., T., Occ. represent Map, Box, Depth, Motion, Tracking, and Occupancy respectively.

• The VAD† means we evaluate VAD with the metrics in Occworld.

Table 2: Quantitative results of motion planning. OCCVAR achieves competitive performance while relying solely on 3D semantic occupancy.

cworld, OCCVAR improves the average IoU from 29.17 to 31.03 and improves the average mIoU from 19.93 to 22.06. For camera input, we observe that OCCVAR improves the average IoU from 22.99 to 24.35 and improves the average mIoU from 8.66 to 11.79. This results highlights the strong predictive performance of OCCVAR, which sets the state-of-the-art on nuScenes val dataset.

**Motion Planning** We compare the motion planning performance of OCCVAR with several strong baselines that utilize various inputs and supervision methods. The results are shown in Tab. 2. We observe that UniAD (Hu et al., 2023b) achieves the best performance among these methods. However it relies heavily on multiple supervisions, including map segmentation, detection, depth estimation, tracking and occupancy prediction. The excessive auxiliary tasks limits its scalability to large-scale datasets. As an alternative, OCCVAR achieves competitive performance while relying solely on 3D semantic occupancy. We can see that the results of OCCVAR are slightly worse than the baselines. We attribute this to the information loss caused by the discretization of ego motion. However, such discretization has a positive impact on generating occupancy scenes.

### 4.3 ABLATION RESULTS

To delve into the effect of each module, we conduct a comprehensive ablation study on OCCVAR.

**Long-term Generation.** To evaluate the long-term generation capabilities of OCCVAR, we conducted a series of experiments comparing its performance against OccWorld (Zheng et al., 2023). As shown in 4, we observe that OccWorld exhibits repetition artifacts when generating long time series. Specifically, after a certain number of time steps, the model begins to produce repetitive patterns, which diminishes the fidelity of the generated 3D scenes.

In contrast, OCCVAR demonstrates significantly improved performance in long-term sequence generation. For example, the geometry of the bus in the first sequence is well-maintained over time. By leveraging next-scale prediction, our model avoids the common pitfalls of next-token autoregressive models. As a result, OCCVAR produces coherent, high-fidelity occupancy scenes that more closely mirror real-world 3D dynamics, even in long-term predictions.

**Effectiveness of Progressive Training.** In next-scale prediction, a reasonable training method can greatly improve the convergence speed of the model. One of the training strategies is the progressive training Tian et al. (2024a). Progressive training in next-scale prediction involves gradually increasing the complexity of the task. The model first learns to predict lower-resolution features before moving on to higher resolutions. This staged approach helps stabilize training and enhances performance in generating detailed visual content. Specifically, we start with training the model on lower-resolution token, allowing it to learn basic structures and patterns. For example, in the first stage, we only calculate the loss of the 0-th token (motion token) to learn the the motion information of ego car. Then we gradually introduce the loss from low resolution to high resolution. The loss of each scale gradually increases in a warm up manner and only proceeds to the next scale after train-

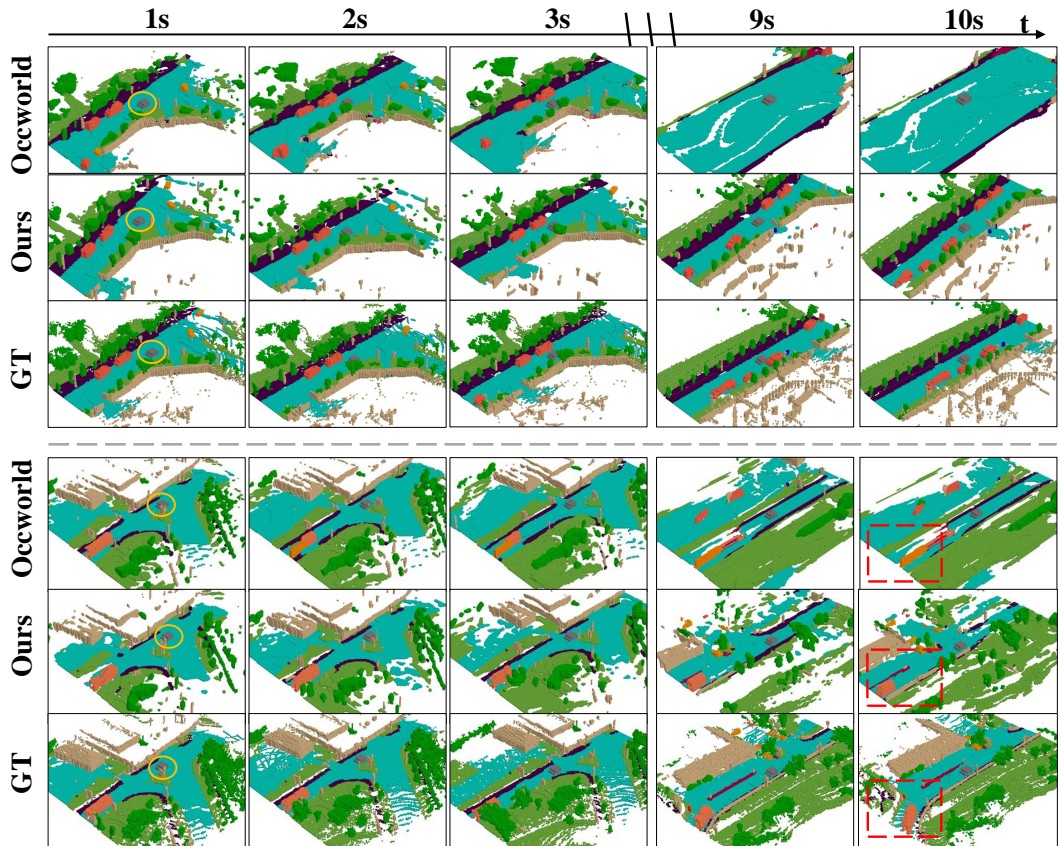

Figure 4: **Qualitative results.** We compare OCCVAR with OccWorld (Zheng et al., 2023) in generating long sequences. OccWorld exhibits repetition artifacts. In contrast, OCCVAR produces more diverse and realistic occupancy scenes. We mark the ego vehicle with an orange circle in the first column.

| Training | MIOU(%)↑ | | | | IOU(%)↑ | | | |
|---|---|---|---|---|---|---|---|---|
| | 1s | 2s | 3s | Avg. | 1s | 2s | 3s | Avg. |
| Direct | 26.83 | 20.92 | 14.95 | 20.90 | 36.74 | 27.10 | 22.09 | 28.64 |
| Progressive | **27.96** | **21.75** | **16.47** | **22.06** | **38.73** | **29.50** | **24.86** | **31.03** |

Table 3: Ablation study of progressive training. Progressive training is a key strategy in next-scale prediction and has a positive impact on training results.

ing to convergence. The results are shown in Tab. 3. We can see that the progressive training can enhance the generation capability of the model greatly, demonstrating the importance of progressive training in OCCVAR. We display the details of progressive training process in Appendix. A.5.

**Efficiency.** The latency is of great significance for the deployment of autonomous driving system. In this experiment, we compare OCCVAR with existing works, including diffusion-based approaches Wang et al. (2024a) and autoregressive approaches Zheng et al. (2023). Since OccLLaMA Wei et al. (2024) is not open source, we do not report their inference time. Moreover, OccWorld is not a standard AR model that utilizes next-token prediction, we also adapt our method for next-token prediction. Specifically, we utilize the tokenizer in Occworld (Zheng et al., 2023) while employing the same generation architecture as ours, denoted as Ours (AR). The results are shown in Tab. 4. We can see that compared with next-token prediction (Ours AR), the next-scale prediction (Ours VAR) has a much lower latency while demonstrating better performance. This indicates the superiorty of the next-scale prediction rather than next-token prediction in 4D occupancy world model. It is worth mentioning that although Occworld also has a very fast inference speed, the OCCVAR outperforms Occworld by 4.9% on MIoU and 4.4% on IoU.

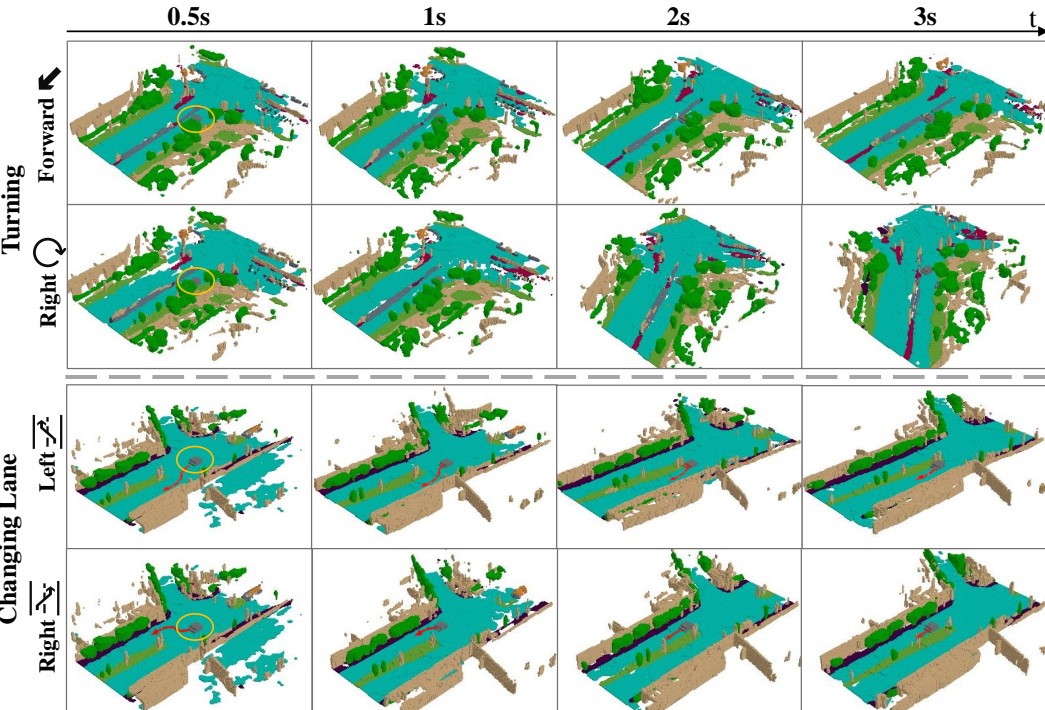

Figure 5: **Qualitative results of motion controllability.** OCCVAR successfully generates results aligned with the motion input, thus enabling precise control of the vehicle's motion like turning (top) or changing lane (bottom). We mark the ego vehicle with an orange circle in the first column.

| Method | Para.(M) | Latency(s) | MIOU(%)↑ | IOU(%)↑ |
|--------|----------|-----------|----------|---------|
| OccSora (Wang et al., 2024a) | 9896 | ∼ 20 | - | - |
| Occworld (Zheng et al., 2023) | 7238 | 0.35 | 17.14 | 26.63 |
| OccLLaMA (Wei et al., 2024) | - | - | 19.93 | 29.17 |
| Ours (AR) | 5591 | ∼ 5 | 19.20 | 26.63 |
| Ours (VAR) | 5881 | 0.56 | 22.06 | 31.03 |

Table 4: Efficiency analysis. Para. refers to the number of parameters. The latency refers to the inference time of generating a scene with 6 frames.

**Motion Controllability** Controllability refers to the model's capacity to precisely adhere to these inputs, ensuring that the generated scenes reflect the specified conditions with high fidelity. Ego-motion control is particularly critical, as it ensures that the model generates scenes from the correct viewpoint and perspective. In this setting, we manipulate the ego-motion inputs and measure the performance of the resulting occpupancy scenes in terms of scene geometry and temporal consistency. As shown in Fig. 5, OCCVAR can generate the corresponding results that collaboratively aligned with the conditional motion input, indicating the powerful generalization ability of our method.

## 5 CONCLUSIONS

In this paper, we present OCCVAR, an innovative autoregressive occupancy world model designed to enhance the efficiency and accuracy of 3D scene prediction for autonomous driving applications. By integrating next-scale prediction and a multi-scale scene tokenizer, OCCVAR effectively captures hierarchical spatial information while maintaining temporal consistency. Our extensive evaluations on the nuScenes dataset demonstrate that OCCVAR surpasses existing methods, achieving improvements of 2.13% in IoU and 1.86% in mIoU, setting new benchmarks in the field. These results highlight the potential of OCCVAR to facilitate real-time applications in autonomous driving, paving the way for future advancements in occupancy modeling.

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

## A APPENDIX

### A.1 TIME COMPLEXITY

In this section, we show the proof of the time complexity of next-token prediction and next-scale prediction generation illustrated in VAR Tian et al. (2024a). Note that the complexity is computed for generating one frame with a standard self-attention transformer. We assume the total number of BEV tokens are $x \times y = n^2$, where $(x, y)$ are the resolution of the BEV feature map and $x = y = n$. For next-token prediction, during the $i$-th ($1 \leq i \leq n^2$) autoregressive iteration, the attention scores are computed between the $i$-th token and all previous tokens, requiring $\mathcal{O}(i^2)$ time. Thus the total time of generating $n^2$ tokens is:

$$\sum_{i=1}^{n^2} i^2 = \frac{1}{6} n^2 (n^2 + 1)(2n^2 + 1), \tag{7}$$

which is equivalent to $\mathcal{O}(n^6)$ basic computation.

For next-scale prediction, assume that we utilize $M$ resolutions $(x_1, y_1, x_2, y_2, ..., x_M, y_M)$ of BEV space and $x_M = y_M = n$. $M$ is a constant. We set $n_m = a^{m-1}$ where $a > 1$ is a constant such that $a^{M-1} = n$. In this assumption, there are $M$ autoregression iterations and $M = \log_a n + 1$.

Consider the $m$-th ($1 \leq m \leq M$) autoregression iteration, the attention scores are computed between the $m$-th scale and all previous scales. The total number of tokens to be attended is:

$$\sum_{i=1}^{m} n_i^2 = \sum_{i=1}^{m} a^{2 \cdot (m-1)} = \frac{a^{2m} - 1}{a^2 - 1} \tag{8}$$

Thus the complexity of the $m$-th autoregression iteration is $(\frac{a^{2m}-1}{a^2-1})^2$. The total time of generating $M$ scale tokens is:

$$\sum_{m=1}^{\log_a(n)+1} \left(\frac{a^{2m}-1}{a^2-1}\right)^2 \tag{9}$$

$$= \frac{(a^4-1)\log n + \left(a^8 n^4 - 2a^6 n^2 - 2a^4(n^2-1) + 2a^2 - 1\right)\log a}{(a^2-1)^3(a^2+1)\log a} \tag{10}$$

$$\sim \mathcal{O}(n^4). \tag{11}$$

## A.2 ATTENTION MASK VISUALIZATION

We show the attention mask of each transformer block in Fig. 6. In temporal attention blocks, we employ a scale-wise causal attention, where each token can attend on its prefix as well as the tokens in the same scale. In spatial attention blocks, we employ an intra-frame full attention, where each token can attend to the tokens in the same frame. In var attention blocks, we employ an inter-frame causal attention, where each token can attend to its prefix as well as the tokens in the same scale. Note that when generating $t+1$-th frame, the var attention blocks only take the tokens in previous frame (instead of all frames) as prefix for efficiency.

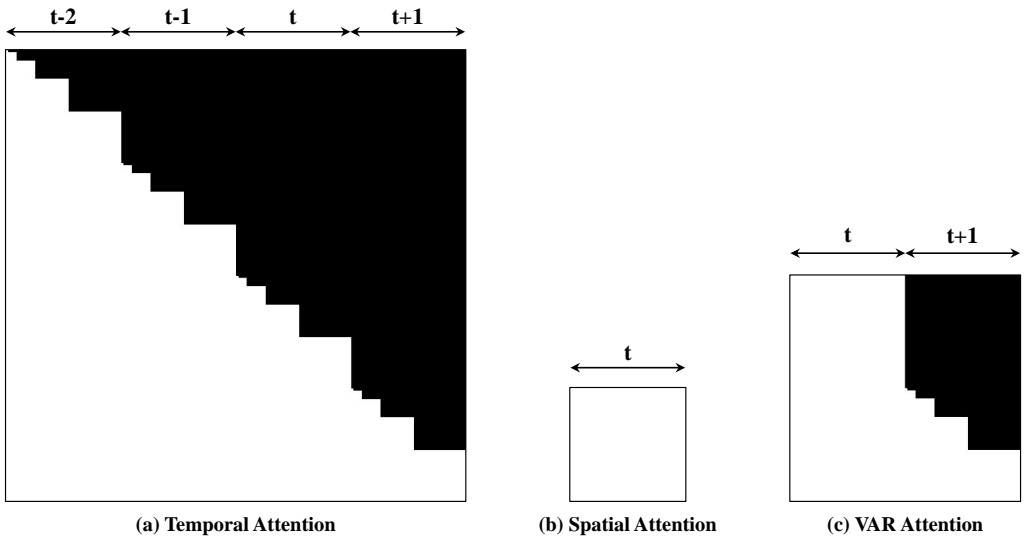

(a) Temporal Attention  (b) Spatial Attention  (c) VAR Attention

Figure 6: Visualization of attention mask of three types of transformer blocks.

## A.3 IMPLEMENTATION DETAILS

We utilized the deepspeed (Rasley et al., 2020) for accelerating training and saving memory. We utilize AdamW optimizer and a Cosine Annealing scheduler for training, where the learning rate is 1e-4 and the weight decay is 0.01. The whole training are conducted on 8 NVIDIA A100.

## A.4 EFFECTIVENESS OF MULTI-SCALE TOKENIZER.

We compare the reconstruction performance of different hyperparameters of the tokenizer, results shown in Tab. 5. We also show the reconstruction performance of the existing methods for comparison. We observe that our multi-scale scene tokenizer outperforms baseline in a large margin, demonstrating the superiority of multi-scale quantization. We also observe that larger resolution of the latent BEV feature map resolution delivers better performance. However, the increase in resolution will also lead to a significant increase in the number of tokens after discretization, which influences the performance in the generation stage. Thus we utilize the BEV with resolution $(25 \times 25)$ in our autoregressive modeling.

| Method | Setting | | | Reconstruction | |
|---|---|---|---|---|---|
| | Res. | Dim. | Size | MIOU(%)↑ | IOU(%)↑ |
| Occworld | 50 | 128 | 512 | 66.38 | 62.29 |
| OccLLaMA | 50 | 256 | 4096 | 70.94 | 61.03 |
| Ours | 25 | 128 | 4096 | **57.83** | **49.76** |
| Ours | 50 | 128 | 4096 | **75.09** | **68.96** |

Table 5: Ablation study of tokenizer parameters.

## A.5 PROGRESSIVE TRAINING

In this section, we show the the process of the progressive training. Progressive training in next-scale prediction involves gradually increasing the complexity of the task. We first train the 0-th scale token (motion token) until convergence. Then we gradually add the loss of 1-th scale to the total loss in a warm-up approach. We utilize a linear warm-up function and the warm-up process for each scale lasts for 10 epochs.