# OpenReview forum: "OccVAR: Scalable 4D Occupancy Prediction via Next-Scale Prediction"
_ICLR.cc/2025/Conference — ICLR 2025 Conference Withdrawn Submission_

### Official Review · Reviewer_UPhr · 2024-10-21

**Soundness:** 2
**Presentation:** 2
**Contribution:** 2
**Rating:** 3
**Confidence:** 5

**Summary:**

The authors proposed a method for occupancy forecasting and long-term occupancy generation. This paper addresses issues of efficiency and temporal degradation. To tackle these challenges, the temporal next-scale prediction is proposed to perform coarse-to-fine autoregressive modeling. Its effectiveness is verified on the Occ3D and nuScenes datasets.

**Strengths:**

This paper discusses two critical issues in the field of occupancy world models: efficiency and long-term generation.

**Weaknesses:**

Overall, this paper is an incremental work with limited contributions and novelty.
1. The novelty in the framework structure is limited. Compared to previous works, OccVAR only adopts a coarse-to-fine strategy and incorporates attention mechanisms to model temporal, spatial, and multi-scale dimensions.
2. In terms of efficiency, OccVAR cannot be regarded as more efficient than its predecessors. Table 4 indicates that its latency is higher than that of OccWorld.
3. Regarding long-term generation, OccVAR does not show any clear advantage over existing methods such as OccSora and DOME.
4. For motion planning, OccVAR's performance is inferior to both OccWorld and UniAD.

**Questions:**

1. Lack of effective explanation for Figure 1. What does "Pose Control" mean? Why does the entire scene rotate in the top-right section of Figure 1?
2. Figure 2 references a "robust tokenizer". How is this robustness demonstrated or reflected?
3. OccSora is also capable of controllable occupancy generation. Why is there no quantitative comparison made with it?
4. In Table 4, only the number of parameters and the inference time for generating a scene are reported. I suggest including a comparison of memory overhead. Additionally, OccVAR shows no advantage in terms of latency. Compared to OccWorld, OccVAR's latency is 1.6 times higher, which does not address the inefficiency problem seen in prior methods.
5. As explained in the paper, OccVAR performs worse than UniAD in motion planning due to issues related to supervision. However, why is OccVAR's motion planning performance worse than OccWorld's?
6. How is the "scalable" reflected in the title of the paper?
7. Why is OccVAR significantly less accurate than DOME [a] in 4D occupancy forecasting? Additionally, DOME can perform long-term generation, up to 32 seconds. Can OccVAR's long-term generation capabilities be comparable to DOME's?

[a] Gu, S., Yin, W., Jin, B., Guo, X., Wang, J., Li, H., ... & Long, X. (2024). DOME: Taming Diffusion Model into High-Fidelity Controllable Occupancy World Model. arXiv preprint arXiv:2410.10429.

---

### Official Review · Reviewer_y7wn · 2024-10-30

**Soundness:** 2
**Presentation:** 2
**Contribution:** 2
**Rating:** 3
**Confidence:** 4

**Summary:**

The paper introduces an autoregressive occupancy world model (OCCVAR) designed for autonomous driving applications. OCCVAR incorporates a spatial-temporal transformer with temporal next-scale prediction to efficiently predict 4D occupancy scenes, capturing both the fine-grained 3D geometry and dynamic evolution of environments. A multi-scale tokenizer is utilized to encode hierarchical scene information, enhancing the model's ability to reconstruct and generate long-time occupancy scenes with high quality. Experiments on the nuScenes dataset demonstrate OCCVAR's superior performance over previous methods, with improvements of 2.13% in IoU and 1.86% in mIoU, showcasing its potential for real-time applications in autonomous driving.

**Strengths:**

1. OCCVAR is capable of scalable 4D occupancy world modeling, which is crucial for simulating the movement of the ego vehicle and the evolution of the surrounding environment over time.
2. The model uses a coarse-to-fine autoregressive mechanism, specifically temporal next-scale prediction, which is effective in handling the inefficiency and spatial degradation problems in previous works.
3. It employs a multi-scale tokenizer designed to capture hierarchical information of 3D scenes, leading to better reconstruction and generation of occupancy scenes.

**Weaknesses:**

1. This paper lacks novelty. It basically applies VAR to OccWorld without substantial improvements. The mutli-scale scene tokenizer is rooted from VAR, and OccWorld already uses spatial and temporal attention.
2. OCCVAR achieves worse planning performance than OccWorld. They should explain more about this rather than a simple description of the discretization of ego motion.
3. They only compare long-term generation with OccWorld, while OccSora also claims to achieve this long-term generation. They should also discuss more about the differences with OccSora and the superiority of their method.
4. While the paper includes some ablation studies, more comprehensive studies could be conducted to understand the contribution of each component of the model. For instance, studying the impact of different numbers of scales in the tokenizer or the effect of varying the training regime could provide further insights.

**Questions:**

1. What is the planning performance if using continuous ego token? Will it outperform OccWorld?
2. Can OCCVAR achieve better long-term generation performance than OccSora?

---

### Official Review · Reviewer_SYMx · 2024-10-30

**Soundness:** 3
**Presentation:** 3
**Contribution:** 3
**Rating:** 6
**Confidence:** 3

**Summary:**

The paper introduces OCCVAR, a generative occupancy world model designed for autonomous driving systems. OCCVAR simulates the movement of the ego vehicle and the evolution of the surrounding environment, capturing fine-grained 3D geometry and dynamic changes in 3D scenes. The model addresses the challenges of inefficiency and temporal degradation faced by previous autoregressive approaches in long-time generation.

**Strengths:**

1. OCCVAR uses a spatial-temporal transformer with temporal next-scale prediction (TNSP) to improve efficiency and quality, predicting 4D occupancy scenes from coarse to fine scales.
2. The model incorporates ego movement into the tokenized occupancy sequence, enabling the prediction of ego movement and controllable scene generation.
3. OCCVAR utilizes a multi-scale tokenizer to capture hierarchical information of 3D scenes, enhancing the modeling of fine-grained 3D geometry.

**Weaknesses:**

1. Although with occupancy groundtruth input, the planning performance of the model is still lower than camera-input methods like UniAD and VAD.
2. The paper does not contain necessary ablation studies to validate the effectiveness of its architectural designs.
3. The efficiency comparison in Table 4 is confusing because I think the generation process of VAR actually includes that of direct AR, but why is the inference time of VAR even shorter than that of AR?
4. The parameter comparison in Table 4 is also confusing because a model for autonomous driving normally contains ~100M parameters.

**Questions:**

1. What could be the reason for the inferior planning performance of OCCVAR compared with models with image input?
2. What is the role of the next scale prediction design in OCCVAR achieving higher mIoU compared with the other methods?
3. Further explanation of Table 4.

---

### Official Review · Reviewer_jyqn · 2024-11-02

**Soundness:** 2
**Presentation:** 2
**Contribution:** 1
**Rating:** 3
**Confidence:** 4

**Summary:**

To address the occupancy prediction task in autonomous driving, the authors propose OCCVAR, which designs a spatiotemporal transformer for temporally next-scale prediction. They also implement a coarse-to-fine scale approach to predict 4D occupancy scenes. Additionally, the authors incorporate ego-motion into the segmented occupancy sequence, enabling the prediction of ego-motion and controllable scene generation.

**Strengths:**

1. The authors model occupancy using a coarse-to-fine approach, achieving more refined 3D reconstruction.

2. The authors propose encoding ego-vehicle motion and integrating it into the occupancy sequence to enhance scene prediction capabilities.

**Weaknesses:**

1. The coarse-to-fine approach proposed by the authors was originally introduced by SurroundOcc [1], and many related works, such as OpenOccupancy [2] and Pyramid Diffusion [3], have also adopted this method, making its novelty limited.

2. The method proposed by the authors shows relatively poor performance both in reconstruction quality and future prediction capabilities compared to other methods. The slight improvement in L2 prediction accuracy at the 1-second mark compared to other methods does not sufficiently demonstrate the effectiveness of the proposed approach.

[1] Wei, Yi, et al. "Surroundocc: Multi-camera 3d occupancy prediction for autonomous driving." Proceedings of the IEEE/CVF International Conference on Computer Vision. 2023.

[2] Wang, Xiaofeng, et al. "Openoccupancy: A large scale benchmark for surrounding semantic occupancy perception" ICCV 2023.

[3] Liu, Yuheng, et al. "Pyramid Diffusion for Fine 3D Large Scene Generation." arXiv preprint arXiv:2311.12085 (2023).

**Questions:**

1. The authors should include an ablation study to demonstrate the impact of each module on the results.

2. The authors should explain the advantages of the coarse-to-fine design for future scene prediction, clarifying why their proposed method performs better than OccWorld [1] and similar methods only in the first step of Quantitative Results of Motion Planning but falls behind in subsequent steps. This contradicts the claim of high accuracy in long-term predictions stated in the title.

[1] Zheng, Wenzhao, et al. "Occworld: Learning a 3d occupancy world model for autonomous driving." arXiv preprint arXiv:2311.16038 (2023).

---

### Note · Authors · 2024-11-15

I have read and agree with the venue's withdrawal policy on behalf of myself and my co-authors.